# Aglianico Grape Seed Semi-Polar Extract Exerts Anticancer Effects by Modulating MDM2 Expression and Metabolic Pathways

**DOI:** 10.3390/cells12020210

**Published:** 2023-01-04

**Authors:** Rossana Cuciniello, Francesco Di Meo, Maria Sulli, Olivia Costantina Demurtas, Mirella Tanori, Mariateresa Mancuso, Clizia Villano, Riccardo Aversano, Domenico Carputo, Alfonso Baldi, Gianfranco Diretto, Stefania Filosa, Stefania Crispi

**Affiliations:** 1Institute of Biosciences and BioResources-UOS Naples CNR, Via P. Castellino 111, 80131 Naples, Italy; 2IRCCS Neuromed, 86077 Pozzilli, Italy; 3Department of Medicine, Indiana University School of Medicine, 975 W Walnut Street, Indianapolis, IN 46202, USA; 4Division of Biotechnology and Agroindustry, Biotechnology Laboratory, ENEA, Casaccia Research Center, Via Anguillarese 301, 00123 Rome, Italy; 5Division of Health Protection Technologies, ENEA, Casaccia Research Center, Via Anguillarese 301, 00123 Rome, Italy; 6Department of Agricultural Sciences, University of Naples Federico II, Via Università 100, 80055 Portici, Italy; 7Department of Environmental, Biological and Pharmaceutical Sciences and Technologies, University of Campania “L. Vanvitelli”, 81055 Caserta, Italy

**Keywords:** grape-seed extract, apoptosis, MDM2, gene expression, cancer metabolism, proanthocyanidins, natural molecules

## Abstract

Grapevine (*Vitis vinifera* L.) seeds are rich in polyphenols including proanthocyanidins, molecules with a variety of biological effects including anticancer action. We have previously reported that the grape seed semi-polar extract of Aglianico cultivar (AGS) was able to induce apoptosis and decrease cancer properties in different mesothelioma cell lines. Concomitantly, this extract resulted in enriched oligomeric proanthocyanidins which might be involved in determining the anticancer activity. Through transcriptomic and metabolomic analyses, we investigated in detail the anticancer pathway induced by AGS. Transcriptomics analysis and functional annotation allowed the identification of the relevant causative genes involved in the apoptotic induction following AGS treatment. Subsequent biological validation strengthened the hypothesis that MDM2 could be the molecular target of AGS and that it could act in both a p53-dependent and independent manner. Finally, AGS significantly inhibited tumor progression in a xenograft mouse model of mesothelioma, confirming also *in vivo* that MDM2 could act as molecular player responsible for the AGS antitumor effect. Our findings indicated that AGS, exerting a pro-apoptotic effect by hindering MDM2 pathway, could represent a novel source of anticancer molecules.

## 1. Introduction

Grapevine (*Vitis vinifera* L.) is a plant rich in bioactive compounds that are principally distributed in berry skin, stem, leaves and seeds. Among the substances accumulated in grapes, the most biologically interesting ones are represented by polyphenols. They constitute the most important class of natural bioactive molecules. Therefore, there is an increasing interest in their use for the treatment and prevention of different chronic pathologies, including cancer. Grape seeds are a significant source of polyphenols, containing about 70% of these compounds accumulated in grapes [1]. In particular, these polyphenols belong mainly to the flavonoids, anthocyanins, proanthocyanidins and flavan-3-ols sub-classes. Among them, proanthocyanidins are known to be the key molecules responsible for a wide range of biological functions including antioxidant, anticancer, anti-inflammatory, antimicrobial, cardioprotective and neuroprotective functions [2]. More in detail, proanthocyanidins are oligomeric flavonoids generated downstream of phenolic acids, chalcones, flavanones and flavonols [3] that can be classified as oligomers (2–10 monomers) or polymers (>10 monomers), depending on the polymerization complexity. In addition, monomers can polymerize to form different procyanidins depending, also, on the connection modes and the monomer types [4]. Oligomers containing 2–10 monomers represent the most bioactive proanthocyanidins. The effectiveness of the oligomers could be also due to the higher bioavailability, since larger proanthocyanidins polymers are scarcely absorbed by the body in comparison to the shorter ones. Increasing evidence indicates that proanthocyanidins exert anticancer properties acting on multiple oncogenic signaling pathways [5,6,7]. These properties are linked to the peculiar ability of specific proanthocyanidins to interact with and modify the activity of proteins involved in the biological activities resulting from their use [7,8]. 

We have recently described the anticancer properties of semi-polar grape seed (GS) extracts from two Campania cultivars, Aglianico (AGS) and Falanghina (FGS), in three mesothelioma cell lines [9] differing in their sensitivity and resistance to the standard combined treatment with cisplatin and piroxicam [10]: MSTO-211H (MSTO), NCI-H2452 (NCI) and Ist-Mes2 (Mes2). Both the extracts resulted to be active in decreasing cancer properties, even if the AGS extracts provided the strongest effects. Subsequent metabolomic analyses indicated a high content in proanthocyanidins, with a maximum number of 11 polymerized units, enriched in AGS in comparison to FGS; this suggested that oligomeric proanthocyanidins might be involved in determining the detected anticancer activity [9]. In the present study, we unraveled the molecular signaling pathway involved in AGS anticancer effects in a mesothelioma tumor by means of multi-omics analysis. We first performed a transcriptomic analysis and a functional annotation with the aim of identifying the relevant causative genes associated with the treatment. Then, to deepen understanding of the AGS ability to modulate cancer cell metabolism, we performed a metabolomic analysis, which highlighted that AGS treatment modulates several metabolic pathways which play a role in cancer progression. Finally, the efficacy of AGS treatment was evaluated *in vivo* using a xenograft mouse model of mesothelioma. Overall, our data shed light on the mechanism, at a molecular-biochemical level, by which AGS exerts anticancer activity and provided evidence that it is able to reduce tumor growth *in vivo*.

## 2. Materials and Methods 

### 2.1. Cell Culture

All the culture media and supplements were purchased from Euroclone (Euroclone SpA, Pero (MI), Italy). Supplements for the growth of MCF10A cells were purchased from Merck (Merck group KGaA, Darmstadt, Germany). Human mesothelioma cell lines MSTO-211H (MSTO) and NCI-H2452 (NCI) were grown in RPMI 1640. Human mesothelioma Ist-Mes2 (Mes2) cells were cultured in DMEM. The human breast adenocarcinoma cell line MDA-MB-231(MDA) was grown in RPMI 1640, while human mammary epithelial cell line, MCF10A, was cultured in DMEM/Ham’s F-12 supplemented with 100 ng/mL cholera toxin, 20 ng/mL epidermal growth factor, 0.01 mg/mL insulin and 500 ng/mL hydrocortisone. Finally, human prostate adenocarcinoma cell line PC-3 (PC3) was grown in DMEM/ F-12 and normal prostate epithelium cell line PNT2 was cultured in RPMI 1640. All culture media were supplemented with 10% FBS, glutamine (2 mM), 1% nonessential amino acids and antibiotics (0.02 IU/mL^−1^ penicillin and 0.02 mg/mL^−1^ streptomycin). All cell lines were obtained from the American Type Culture Collection, except for Mes2 cells which were obtained from the Istituto Nazionale per la Ricerca sul Cancro-Genova (Italy). All the experiments reported below were performed in biological triplicate.

### 2.2. AGS Extract Preparation

Aglianico grape seed semi-polar extract (AGS) was prepared following the procedure previously described [9]. Seeds were ground with liquid nitrogen and then lyophilized. The powder was then homogenized with methanol/chloroform (1:4) using TissueLyser (Qiagen, Hilden, Germany). The alcoholic phase was then recovered after a 5′ centrifugation (12,000 rpm at 4 °C) and lyophilized again. Finally, the powder was dissolved in DMSO (Sigma-aldrich, St. Louis, MI, USA) at a concentration of 500 mg/mL and stored at −20 °C.

### 2.3. Cell Proliferation Assay

Approximately 1.5 × 10^5^ cells were treated with 150 or 350 μg/mL of AGS or vehicle (0.1% DMSO) for 24 + 24 h. Cells treated with vehicle were used as the control. The treatment 24 + 24 h was performed as follows: cells were treated twice, 16 h after seeding and again 24 h after the first administration. Subsequently, cells were collected and counted with Trypan Blue solution (Sigma-Aldrich, St. Louis, MI, USA). Cell viability was also evaluated by counting live cells using MTS assay (CellTiter 96; Promega Corporation, Madison, WI, USA) according to the manufacturers’ instructions. For the MTS assay, the absorbance was recorded on a microplate reader at a wavelength of 490 nm (VICTOR Multilabel Plate Reader; PerkinElmer, Inc., Waltham, MA, USA). All experiments were performed in biological triplicate and data are expressed as the mean ± SD. 

### 2.4. Apoptosis Detection by Flow Cytometry Analysis

Approximately 7.5 × 10^5^ cells were plated in 100 mm plates. After overnight incubation, cells were treated with 350 μg/mL of AGS extract for 24 + 24 h and stained with Propidium Iodide and Annexin V (Annexin V FITC assay, BD Biosciences, Franklin Lakes, NJ, USA), according to the manufacturer’s guidelines. Flow cytometry was performed using a flow cytometry system (FACSCanto, BD Biosciences Franklin Lakes, NJ, USA). All experiments were performed in biological triplicate and data are expressed as the mean ± SD. 

### 2.5. Transcriptomic Analysis

MSTO-211H (1.5 × 10^6^) were treated with 350 μg/mL of AGS for 24 + 24 h and RNA was then extracted using Trizol (Life Technologies, Carlsbad, CA, USA) and following manufacturer’s instructions. Total RNA samples were quantified with a spectrophotometer (NanoDrop 1000, Thermo Fisher Scientific, Waltham, MA, USA) and the quality was assessed by Bioanalyzer analysis (Agilent, Santa Clara, CA, USA).

Expression profiling was performed using 100 ng RNA and the Affymetrix Human Clariom S Assay (Affymetrix/Thermo Fisher Scientific, Waltham, MA, USA), which interrogates over 20,000 well-annotated genes. Total RNA was converted in labelled ssDNA and used to hybridize the microarray. All the procedures, including washing and staining of the array, were performed following Affymetrix instructions. After laser scanning, CEL files were used for data analysis. Primary data analysis was performed with the TAC 4.0 (Transcriptome Analysis Console software version 3.1.0.5-Thermo Fisher Scientific, Waltham, MA, USA) and the quality was assessed by Bioanalyzer analysis (Agilent, Santa Clara, CA, USA) including the Robust Multiarray Average module for normalization. Transcriptional modifications were detected, comparing AGS treated cells to untreated cells. Differentially expressed genes were selected applying as a cut-off criteria log2-fold change ≥ + /−2 and a *p* value ≤ 0.05.

Microarray data were deposited to GEO repository with the accession number GSE208183.

### 2.6. IPA Analysis

Functional annotation, pathway and network analysis as well Bioprofiler analysis were performed using Qiagen IPA software (https://digitalinsights.qiagen.com/, accessed on 8 August 2022). The “core analysis” function included in the software was used to interpret the differentially expressed data, which included biological processes, canonical pathways, upstream transcriptional regulators and gene networks. The BioProfiler tool includes all the scientific information to generate molecular profiles of diseases, phenotypes and biological processes, listing all the genes and compounds that have been associated with the profiled term.

### 2.7. Quantitative PCR

To validate expression of genes selected by transcriptomic analysis, RNA was extracted using Trizol (Life Technologies, Carlsbad, CA, USA) following the manufacturer’s instructions. For qPCR (quantitative polymerase chain reaction) assays, 100 ng of total RNA from each AGS-treated and untreated sample were retro-transcribed using the High Capacity cDNA Reverse Transcription Kit (Applied Biosystem, Waltham, MA, USA). Then, qPCR reactions were performed using 7900 HT Real Time PCR (Applied Biosystem). Gene specific primers for the selected genes are reported in Table 1. 

Primers were designed at exon–exon junctions using Primer express 2.0 (Applied Biosystems, Whaltam, MA, USA). Target expression level was performed as previously described [11], using *GAPDH* as a housekeeping gene. All the experiments were performed in biological triplicate. 

### 2.8. Western Blot

Protein extracts were prepared as previously described [10]. For each lane, 20 μg of total cell lysates were separated in 4–15% Tris–glycine gels (Bio-Rad Laboratories, Inc., Hercules, CA, USA) at 100 V. Proteins were then transferred to 0.2 μm PVDF membranes (Bio-Rad Laboratories, Inc.), probed with the specific primary antibodies, followed by secondary antibodies conjugated with horseradish peroxidase according to manufacturer’s indications. Primary antibodies used for Western blot include MDM2 (Santa Cruz Biotechnology, #sc-965, Santa Cruz, CA, USA), Phospho-c-Jun (Ser243) (Cell Signaling, #2994, Danvers, MA, USA), p53 (Cell Signaling, #2524), p21^Waf1/Cip1^ (Cell Signaling, #2947), p27^Kip1^ (BD Biosciences, #K25020). β-Actin (Cell Signaling, #3700) was used as the loading control. All the antibodies were used at working concentration indicated by manufacturers. Protein bands were detected by Clarity western ECL (Bio-Rad Laboratories, Inc.) and the blots were semi-quantified with ImageJ software. All the experiments were performed in biological triplicate.

### 2.9. Metabolomics Analysis

LC-HESI-HRMS analysis of the control and treated cell line metabolome was performed as previously described [12,13]. Semi-polar metabolites were extracted from 1 × 10^6^ cells that were lysed on ice by sonication at 10 Hz output (3 × 10 s) and homogeneously ground with 0.75 mL of cold 75% (*v*/*v*) methanol, 0.1% (*v*/*v*) formic acid, spiked with 0.5 μg/mL formononetin as an internal standard. After shaking for 40′ at 20 Hz using a Mixer Mill 300 (Qiagen), samples were centrifuged for 20 min at 20,000× *g* at 4 °C. Then, 0.6 mL of supernatant were removed and transferred into filter (PTFE) Whatman^®^ vials for LC/MS analysis (Sigma-Aldrich, Darmstadt, Germany). For each experimental group and for each cell line analyzed, 3 independent biological replicates, consisting of 1 × 10^6^ cells were analyzed for both AGS treated and untreated cells. For each biological replicate, at least one technical replicate was carried out. Liquid chromatography (LC) was carried out using a Phenomenex C18 Luna column (100 × 2.0 mm, 2.5 μm; Phenomenex, Torrance, CA, United States) and the mobile phase was composed of water–0.1% formic acid (A) and acetonitrile–0.1% formic acid (B). Total run time was 32 min, and the gradient was 0 to 0.5 min 95% A/5% B, followed by a 24 min linear gradient to 75% B and 26 min 95% A/5% B. 

Five microliters of each sample were injected and a flow of 0.25 was used throughout the LC semi-polar runs. Mass spectrometry analysis was performed using a quadrupole-Orbitrap Q-exactive system (Thermo Fisher Scientific, Cambridge, MA, USA), operating in positive/negative heated electrospray ionization (HESI) coupled to an Ultimate HPLC-DAD system (Dionex, Sunnyvale, CA, USA). Mass spectrometer parameters were set as follows: capillary and vaporizer temperatures 250 °C and 270 °C, respectively, discharge current 5.0 µA, probe heater temperature at 330 °C, S-lens RF level at 50 V. The acquisition was carried out in the 110/1600 m/z scan range, with the following parameters: resolution 70,000, microscan 1, AGC target 1e6 and maximum injection time 50 ms. Full scan MS with data-dependent MS/MS fragmentation was used for metabolite identification. All solvents used were LC-MS grade quality (CHROMASOLV^®^ from Sigma-Aldrich). Metabolites were quantified in a relative way by normalization on the internal standard (formononetin) amounts.

Untargeted metabolomics was carried out as previously reported [9]. Targeted metabolite identification was performed by comparing chromatographic and spectral properties with authentic standards (if available) and reference spectra, in house database, literature data and on the basis of the m/z accurate masses, as reported in the Pubchem database (http://pubchem.ncbi.nlm.nih.gov/, accessed on 8 August 2022) for monoisotopic mass identification or on the Metabolomics Fiehn Lab Mass Spectrometry Adduct Calculator (http://fiehnlab.ucdavis.edu/staff/kind/Metabolomics/MS-Adduct-Calculator/, accessed on 8 August 2022) in the case of adduct detection. 

All the experiments were performed in biological triplicate.

### 2.10. In Vivo Xenograft Model

3 × 10^6^ MSTO cells were resuspended in 100 μL of Matrigel (BD Biosciences) and injected in subcutis (s.c.) of the flank of 6-week-old NU/NU CD1 male mice (Charles River Laboratories, Calco (LC), Italy). Animals were housed in sterilized filter-topped cages kept in laminar flow isolators, fed with autoclaved food and water ad libitum and maintained in a 12 h light/dark cycle. When tumor masses reached the volume of 200–300 mm^3^, mice were enrolled in the study and randomized into two experimental groups: control (Vehicle, 0.1% DMSO *n* = 6) and treated with AGS (100 mg/Kg in 0.1% DMSO, *n* = 8). For each treatment, 500 μL of vehicle alone or AGS extract were administered via intraperitoneal (i.p.) injection three times a week for four weeks. During the experiment, tumors were measured three times a week using a caliper and tumor volume was determined using the formula: (length × width^2^)/2 and body weight registered. The general health status of mice was also constantly monitored. On day 28, mice were euthanized by cervical dislocation and tumor masses were collected for further analyses. 

The percentage of tumor growth inhibition (TGI) was calculated as follows: TGI (%) = (Vc − Vt)/(Vc − Vo) × 100, where Vc and Vt are the median tumor mass of control and treated groups, respectively, at the end of the study and Vo is the volume at the start. 

All the experiments involving animal studies were performed according to the European Community Council Directive 2010/63/EU, approved by the local Ethical Committee for Animal Experiments of the ENEA and authorized by the Italian Ministry of Health (n415/2021-PR).

### 2.11. Histology and Immunohistochemistry

At the end of the treatments, the mice were sacrificed, and tumor specimens assayed for Ki67, for apoptotic index (TUNEL assay) and for MDM2 expression. Immunohistochemistry was performed as previously described [14]. Briefly, tissue samples were fixed in formalin and embedded in paraffin, then sections from each specimen were cut at 3–5 µm, mounted on glass and dried overnight at 37 °C. They were then deparaffinized in xylene, rehydrated through a graded alcohol series and washed in PBS, which was used for all the subsequent washes as well as the antibody dilution. For immunohistochemistry, tissue sections were heated two times in a microwave oven for 5 min at 700 W in citrate buffer (pH 6). Rabbit anti-human Ki67 (DAKO Agilent) and rabbit anti-MDM2 (Elabscience, Houston, TX, USA) were used at final dilution 1:100 for 1 h at room temperature. Then, sections were incubated with UltraTek HRP secondary antibody (ScyTek Laboratories, Logan, UT, USA) for 1 h at room temperature. Diaminobenzidine (ScyTek Laboratories) was used as the final chromogen and hematoxylin was used as a contrast agent. For each tissue section, negative and positive controls were performed, either leaving out the primary antibody or using tissue expressing the antigen of interest. TUNEL reaction was performed using the peroxidases-based Apoptag kit (Sigma-Aldrich, Darmstadt, Germany). TUNEL-positive cells were detected with diaminobenzidine and H_2_O_2_, according to the supplier’s instructions. 

For each specimen, the expression of Ki67 and MDM2, as well as the apoptotic index were evaluated by estimating the number of positive cells visible for high-power field X40 (at least 20 different fields were analyzed for each specimen). The expression for Ki67 was evaluated considering the percentage of positive nuclei, while for MDM2 a score based on percentage of nuclei staining positive was defined as follows: 0 = 0%, 1 = 1–25%, 2 = 26–50% and 3 > 50%. Finally, apoptotic index was estimated counting the number of positive cells for field. All slides were evaluated by the two independent observers (AB and SC) and discordant cases were reevaluated collegially. 

### 2.12. Statistical Analysis

Graph Pad Prism 9.0 (GraphPad Software, Inc., San Diego, CA, USA) analysis was used to evaluate the differences between controls and treatment groups. One-way ANOVA was used to evaluate the significance of the differences among means. Dunnett’s multiple comparison test with Bonferroni post hoc correction was used to assess the significance between each treatment group and the control group. *p* ≤ 0.05 was considered to indicate a statistically significant difference.

As far as metabolomic analysis is concerned, data were statistically validated using an ANOVA plus Tukey’s pairwise t-test comparisons using PAST 4.09 https://www.nhm.uio.no/english/research/infrastructure/past/ (accessed on 8 August 2022). The test was used to identify, for each cell line, the differentially accumulated metabolites (DAMs) in the comparisons between AGS-treated vs. untreated samples. Heatmap and clustering visualization were performed using Morpheus, as previously reported [15]. The Venn diagram was realized using the Venny2.1 tool “https://bioinfogp.cnb.csic.es/tools/venny/index.html” (accessed on 22 August 2022), whereas functional metabolite classification was achieved by through the Metaboanalyst online tool “https://www.metaboanalyst.ca” (accessed on 13 October 2022). For *in vivo* experiments, fold-change in body weight was calculated as the mean percentage value of body weight at day of treatment with respect to the initial value (day 1) for each mouse. 

In regards to *in vivo* analysis, tumor growth data were elaborated as temporal dynamics and statistically analyzed by linear regression curve fit (95% CI; best fit value with 2 parameters: y-intercept and slope) using GraphPad Prism software v9.0 (GraphPad Software, San Diego, CA, USA). Data are expressed as means ± SEM for each group.

## 3. Results

### 3.1. Transcriptomics Profiling 

To evaluate the molecular mechanism of AGS extract in regulating apoptosis in mesothelioma, we performed a transcriptomic analysis using the Clariom S array with the aim to identify the mostly affected pathways. In general, the analysis retrieved a total of 665 genes upregulated and 397 downregulated (Figure 1A and Appendix A).

To functionally annotate the deregulated genes and to find relationships among transcripts specifically involved in apoptosis, we performed IPA (Ingenuity Pathway Analysis) analysis with the purpose of highlighting the major molecular pathways involved in the response to AGS treatment. Notably, as shown in the Figure 1B, the “Cholesterol Biosynthesis” resulted in the most enriched canonical pathway with the highest-ranking signaling pathway (−log *p*-value 2.5; Z-score > 2). Interestingly, cholesterol has been reported to have a cancer-type specific role in cancer development [16]. Indeed, among “Diseases and Disorders”, “Cancer” was the most representative category, and the majority of deregulated genes were associated to cancer, cell death, proliferation and cell viability (Figure 1C). Then, to further analyze the putative target genes associated with AGS treatment, we carried out a Bioprofiler analysis. This approach can generate molecular profiles of diseases or of biological processes by listing the genes associated with the profiled term. This analysis detected *MDM2* (Murine Double Minute 2), an oncogene that acts as a transcriptional regulator, as the causative relevant gene linked to the AGS-induced apoptosis. As shown in Table 2, MDM2 activity, when decreased, is found to be associated with increased apoptosis and cell death. Instead, its increased activity is often associated with cancer traits.

A functional network analysis was performed to evaluate the presence of a functional connection among cholesterol biosynthesis and genes involved in apoptosis. This analysis highlighted a relationship among the genes linked to apoptosis *p53*, *MDM2* and *JUN* (Jun proto-oncogene) and the following transcripts involved in cholesterol metabolism: *HMGCS* (3-hydroxy-3-methylglutaryl-CoA synthase 1), *HMGCR* (3-hydroxy-3-methylglutaryl-CoA reductase), *DHCR24* (24-dehydrocholesterol reductase), *DHCR7* (7-dehydrocholesterol reductase), *FDFT1* (farnesyl-diphosphate farnesyltransferase 1). 

As evidenced in the network, DHCR24 was the only one directly connected to p53 and MDM2 (Figure 1D). Indeed, DHCR24 has been previously reported to reduce the activity of p53, directly by reducing its activation and indirectly by increasing the interaction between MDM2 and p53, thus leading to the ubiquitination of the latter [17]. Additionally, DHCR24 has been shown to interact with MDM2 independently of p53, possibly affecting other MDM2 targets [18]. Notably, the bioinformatic analysis indicated that AGS activates apoptosis in cancer cells targeting MDM2, which has been shown to play oncogenic roles in human cancers and to represent a valuable target for the development of novel cancer therapeutic agents [19].

### 3.2. Biological Validation of Gene and Protein Expression 

To independently validate the expression of the genes highlighted in the functional network, we used q-PCR. More in detail, this analysis was carried out after the treatment with AGS or with the vehicle in all three mesothelioma cell lines: MSTO, NCI and Mes2. Overall, gene expression data confirmed the transcriptomic findings. In fact, we observed a strong downregulation of all the transcripts involved in cholesterol biosynthesis and a modulation of the genes involved in the apoptotic pathway. In particular, after AGS treatment, we found a consistent downregulation of *JUN* and *MDM2*, together with the upregulation of *p53* in all the analyzed cell lines (Figure 2A).

Western blot analysis also confirmed, in all the three mesothelioma cell lines, a decreased expression of MDM2 and of the active form of JUN (*p*-c-JUN) proteins as well as an increased level of p53 (Figure 2B), thus confirming the induction of apoptotic pathway by AGS. 

We also analyzed the protein expression of DHCR24, since it was connected in the network with p53 and MDM2 and its expression has been widely associated with cancer [20,21,22]. However, protein analysis revealed an increase in DHCR24 levels after AGS treatment, thus suggesting that the cholesterol pathway is not related to apoptosis. Despite the increased expression of DHCR24 protein level, we did not observe a reduction in the expression of p53, suggesting that in these cell lines, DHCR24 could be post-translationally regulated as previously reported [23].

These results strengthen the hypothesis that MDM2 could represent the main target of AGS treatment and that its downregulation could play a central role in determining apoptosis. 

### 3.3. AGS Induces Apoptosis in p53 Defective Cancer Cell Lines 

To ascertain if AGS could induce apoptosis through MDM2, we decided to analyze its efficacy in p53 defective cancer cells.

To this aim, we selected two cell lines of breast and prostate cancer characterized by high aggressive behavior and invasiveness: MDA and PC3, respectively. MDA cells belong to the triple-negative breast cancer (TNBC) cells, lacking estrogen and progesterone receptors, as well as HER2 (human epidermal growth factor receptor 2) [24]. They also lack functional p53, due to a missense non-functional mutation [25]. PC3 cells do not express androgen receptor and prostate-specific antigen (PSA) and have a *p53* mutated gene, due to the frame-shift mutation that leads to a truncated unfunctional protein of 168 amino acids [26].

First of all, we analyzed the AGS effect on cell viability on both cell lines. Cells were treated with 150 and 350 μg/mL of AGS concentration in a double treatment for 24 + 24 h (for more details, see methods). In addition, the same treatment was performed on the corresponding non-tumor cells MCF-10 and PNT2, respectively, to verify that AGS cytotoxicity was specific for cancer cells. As shown in Figure 3A, AGS extract was able to reduce cell viability by about 50% compared to untreated cells, in both breast and prostate cancer cells after the double treatment with 350 μg/mL. This result agrees well with our previous findings observed in mesothelioma cell lines [9]. On the contrary, no effect was detected in non-tumor cells MCF10A and PNT2.

To determine if the reduction in viability in these cell lines was caused by apoptosis, we performed a cytofluorimetric analysis. As reported in Figure 3B, AGS treatment determined apoptosis in both cell lines with a stronger effect in breast cancer. In particular, AGS determined an apoptotic increase of about 52% in MDA and about 23% in PC3, compared to the untreated cell.

We also investigated the molecular signaling pathway that triggered apoptosis in the absence of a functional p53. To this aim, we first checked, by q-PCR, the expression of genes included in the functional network (Figure 1D). We verified that in both cell lines *MDM2* and *JUN* followed the same expression pattern previously observed in mesothelioma cells. These results suggest an active role of this pathway in the apoptosis induction, also in the absence of a functional p53.

On the contrary, the analysis of the cholesterol biosynthesis genes was not concordant with the results observed in mesothelioma cells, again indicating that the cholesterol pathway does not account for apoptosis induced by AGS (Figure 4A).

In agreement with the gene expression data, protein levels of MDM2 and JUN clearly showed a decreased amount in both cell lines (Figure 4B). Moreover, as previously reported in different studies, we found unchanged protein expression of p53 in MDA [27], while it was absent in PC3, as expected [28]. These results indicate that AGS extracts are able induce apoptosis by downregulating MDM2, also in absence of functional p53.

To further dissect the signaling pathway involved in MDA and PC3 apoptosis, we focused our attention on two MDM2 molecular targets by analyzing the protein level of two CDKi (Cyclin-dependent Kinase inhibitor): p27 (CDKN1B) and p21(CDKN1A). p27 is a protein downstream of MDM2 that acts as a tumor suppressor [29], while p21 is a protein able to bind MDM2, thus increasing antitumor activity and apoptosis [30,31]. 

Of relevance, the results showed that AGS treatment induced p27 protein upregulation in MDA, while p21 expression was increased in both cell lines (Figure 4B). In breast cancer cells, p53 defective MDM2 inhibition is linked to an increased expression of p27 that partly mediates growth inhibition [32]. Otherwise, in prostate cancer cells, p27 was not modulated, suggesting that it might not be involved in the AGS-induced apoptotic pathway. 

Upregulation of p21 associated with MDM2 downregulation has been linked to apoptosis induction in either prostate and breast cancer cells [33,34]. Overall, these results indicate that AGS is able to induce apoptosis acting on MDM2 and independently from the active p53 protein.

### 3.4. Metabolomic Analysis

To investigate to which extent AGS treatment alters energy metabolism, we investigated, by LC-HRMS analysis, the effect of the AGS extracts on MSTO, PC3 and MDA cells, comparing AGS-treated samples vs. untreated control samples. First of all, we evaluated global metabolomics alterations by untargeted analysis (Appendix A); overall, we found 322 differentially accumulated metabolites (DAMSs) in the AGS-treated samples. Principal component analysis (PCA) was exploited to achieve a general picture of the different samples; as evidenced in Appendix A, untreated and AGS-treated MSTO cells were located closer in the PCA plot, while both PC3 and MDA AGS-treated cells displayed a higher extent of changes when compared to the untreated cells. Subsequently, we used a tentative annotation pipeline for compounds being over- or under-accumulated in at least two out of three lines (Appendix A). More in detail, 25 out of 115 DAMs were identified as increased in all comparisons and included amino acids and dipeptides (homolanthionine, glutamyl-threonine and asparaginyl-threonine). Other members of this class resulted in higher levels in MSTO and PC3 (N-acryloylglycine and N-acetyl-DL-serine), while L-carnitine was reduced in MSTO and MDA. On the contrary, trimethyllysine was higher in PC3 and MDA cells, while valine and 4-amino butanoic acid were reduced in MSTO and MDA [35].

In addition, several fatty acids (margaric acid, (+)-16-methyl stearic acid) and phospholipids such as glycerophosphocholine, (PC(2:0/O-16:0)[U], PC(O-14:0/2:0), PC(2:0/O-16:0)[U], PC(O-14:0/2:0), PE(18:0/0:0), PC(O-16:1(11Z)/2:0) and PC(0:0/18:1(9Z) were positively regulated in all AGS-treated cells. Notably, Alpha-CEHC, a tocopherol catabolite, was increased in all cell lines. Finally, for the additional 12 ions tentative chemical formulas were assigned, although it was not possible to retrieve an unequivocal identification. 

A targeted metabolomics approach allowed us to identify, in AGS-treated samples, 87 compounds involved in cell metabolism. Very interestingly, we found among them molecules belonging to primary pathways involved in cancer metabolism such as glycolysis and the acid citric cycle (tricarboxylic acid, TCA) (Appendix A).

These compounds were used to perform a hierarchical clustering (HCL) analysis in order to evaluate the metabolite profile changes determined by AGS treatment (Figure 5). Metabolites sharing the same profile probably derive from the same biosynthetic pathway or can be part of the same regulatory system. 

As shown in Figure 5, the MSTO AGS-treated and untreated samples are clustered together, suggesting a more similar pattern of response to the AGS treatment; whereas, MDA AGS-treated are placed farthest, thus indicating that they are characterized by the largest and more divergent extent of metabolic changes. Finally, PC3 displayed an intermediate behavior between the two other cell lines under study. 

A more detailed analysis of the metabolic changes occurring in each cell line provided evidence that AGS treatment induced a strong modulation in different metabolic classes. In particular, and in agreement with gene expression data, we found several compounds involved in cholesterol metabolism and catabolism, but the majority of them did not change in the AGS-treated samples compared to control. However, other compounds were found to be decreased after AGS treatment. They included molecules involved in cholesterol biosynthesis. For instance, 14-demethyllanosterol/ 20-alpha,22beta-dihydroxycholesterol decreased in MSTO and PC3, while 4alpha-methylzymosterol-4-carboxylate/Calcitetrol decreased in all cell lines (Figure 5). Additional changes were found in metabolites participating in cholesterol catabolism such as cholesterol alpha-epoxide (5, 6 alpha-epoxy-5-alpha-cholestan-3 beta-ol) being over-accumulated in both MSTO and PC3 and 3-alpha,7alpha,26-trihydroxy-5 beta-cholestan), and over-accumulated in MSTO and MDA [36] cells. 

Of relevance, all the AGS-treated cell lines cells were characterized by further alterations in non-polar and semi-polar metabolic fractions; notable examples of the former are a group of fatty acids which were increased in PC3 and MDA AGS-treated cells (11,12,19-/11,12,20-trihydroxy-5,8,14-eicosatrienoic acid, 14,15,19-/14,15,20-trihydroxy-5,8,11-eicosatrienoic acid) or in MSTO and PC3 (cis-9,10-epoxyoctadecanoic acid (cis-EODA). In addition, levels of 13-keto-9Z,11E-octadecadienoic acid and 14,15-dihydroxy-5,-8,11-eicosatrienoic acid were higher in treated cells (Figure 5). 

Additional changes were found in several compounds of interest: folic acid increased in all cell lines; glucose/fructose and their phosphate forms and two other intermediates in the glycolysis pathway, phosphoenolpyruvate (PEP) and 2,3-phosphoglyceric acid, showed the same positive sign in PC3 and MDA; glyceraldehyde-3-phosphate (GAP) decreased in all treated cell lines. The same trend was observed in both MSTO and MDA cells for 6-O-phosphono-D-gluconic acid and 6-phosphonoglucono-D-lactone, which belongs to the pentose phosphate pathway (PPP), as well as for citric acid and its isoform (isocitric acid) which represent fundamental intermediates of the TCA cycle. Finally, AGS treatment determined the differential accumulation of other groups of interest, such as molecules belonging to the N compound group. For example, AMP was reduced in PC3 and MDA, while thymidine was reduced in MSTO and PC3. Among organic acids, we found benzoic acid to be over-accumulated in all lines and 4-hydroxy and phosphoric acid over-accumulated in PC3 and MDA. On the contrary, lactic acid was reduced in all cells analyzed.

Subsequently, a Venn diagram was used to determine the number of common and specific DAMs within the three cell lines analyzed (Appendix A). In agreement with HCL, MSTO cells treated with AGS exhibited the lowest number of DAMs, with PC3 and MDA sharing 19 common metabolites and a similar number of DAMs (46 and 45, respectively). Most of the common alterations displayed the same sign, with few exceptions as for glyceric acid, over- accumulated in MSTO and PC3 and down-accumulated in MDA; erythrose-4-phosphate, reduced in MSTO and increased in MDA; malic acid, whose amounts were lower in PC3 and higher in MDA cells; and alanine and isoleucine, increased in PC3 and decreased in MDA cells (Table 3). 

However, the analysis revealed only a partial overlap among the different comparisons, with ten metabolites that changed in all cell lines after AGS treatment (Appendix A).

Finally, to highlight the main metabolic pathways affected by AGS treatment, a functional enrichment analysis, performed using the Metaboanalyst tool, was carried out, considering the list of all differentially accumulated compounds. Interestingly, a strong impact on glucose/energy metabolism and more specifically on the Warburg effect, gluconeogenesis and pyruvate metabolism/glycolysis was observed following AGS treatment. Additionally, glutamate and glutathione metabolism as well as carnitine synthesis resulted as the top enriched pathways in AGS-treated cells (Figure 6).

### 3.5. In Vivo Analysis of AGS Treatment

On the basis of results obtained by transcriptomics and metabolome analyses, we decided to assess the efficacy of AGS in reducing *in vivo* tumor growth. To this aim, we set up an experiment using an ectopic xenograft tumor in a mouse model using MSTO cells. The experimental design is reported in Figure 7A. 

The results showed that the administration of AGS (100 mg/kg) three times a week for four weeks was able to significantly delay the growth of the ectopic tumors as compared to the untreated control group (*p*-value < 0.0001). As reported in the Figure 7B, AGS treatment was immediately effective, resulting in a constant and progressive delay in tumor mass. In particular, in the AGS treated mice, the final value of tumor mass was reduced by 40%, confirming an effective tumor growth inhibition in the AGS-treated tumors compared with the untreated control group (*p* < 0.0001; Figure 7B). Importantly, mice treated up to 28 days were healthy and displayed normal behavior and posture. Furthermore, as shown in Figure 7C, no significant variation was observed in body weight of animals treated with AGS with respect to controls at each evaluated time point, suggesting that in our experimental conditions, the grape seed extract is well tolerated.

With the intent to establish whether the inhibitory effect on tumor growth of AGS treatment was related to cell proliferation and apoptosis, Ki67 and TUNEL scores were performed at the end of treatment. Statistical analysis of the obtained scores showed that the proliferation index was significantly lower in tumors of mice treated with AGS, compared to the control group (*p* < 0.0004). Accordingly, the apoptotic index was significantly higher in tumors of mice treated with AGS (*p* < 0.0004). Furthermore, the immunohistochemical analysis of MDM2 expression in the tumors revealed a strong decreased expression in AGS treated samples with respect to control (*p* < 0.0001) (Figure 8). 

These results indicate that AGS treatment acts on both proliferation and apoptosis. Moreover, they confirmed *in vivo* that MDM2 can be considered as one of the molecular players responsible for the antitumor effect of AGS.

## 4. Discussion

Cancer represents the second leading cause of death all over the world and nowadays different therapeutic strategies are available to treat various cancers. However, the use of conventional drugs results in severe side effects due to their toxicity towards healthy cells. Another adverse aspect about the use of standard chemotherapeutics is that their efficacy is often reduced due to the development of cancer resistance—a multifaceted phenomenon related to cancer heterogeneity and occurring through DNA mutations, metabolic changes and drug inhibition or degradation [37]. In the latest decades, increasing research interests have been focused on the development of novel anticancer molecules. Most of the alternative anticancer therapies are based on the use of bioactive molecules extracted from natural sources, which are able to significantly reduce tumor progression and improve healing and survival [38]. In this perspective, several natural molecules have been reported to be non-toxic and effective against several cancers [39]. Our group has recently described the anticancer properties of semi-polar extracts from seed tissues of Aglianico—an Italian grape variety—against mesothelioma, a rare and aggressive cancer developing from the mesothelial surface of the pleural space and characterized by a late diagnosis and lack of effective treatment. In particular, we have reported that AGS in different mesothelioma cell lines was able to affect tumorigenic properties and to induce apoptosis in a time- and dose-dependent manner [9]. In addition, metabolomic analyses indicated that AGS extract was highly enriched in proanthocyanidins, molecules with a maximum number of 11 polymerized units, that could account for the anticancer activity.

In this study, we further dissected, at the molecular level, by means of transcriptomic and metabolomic analyses, the anticancer activity of AGS to find key players responsible for its effect. Deregulated genes were functionally annotated in IPA and this analysis highlighted that, among “Disease and disorders” categories, “Cancer” was indicated as the class containing the majority of genes deregulated by AGS treatment, thus confirming our previous observations. Subsequently, we performed an IPA Bioprofiler analysis that allowed us to identify genes associated with a specific molecular profile of a biological process. This analysis detected MDM2, which was downregulated, as the relevant causative gene linked to the apoptosis induced by AGS treatment.

The successive network analysis depicted an interaction between MDM2 and the other two genes involved in apoptosis: p53 and JUN, which were, respectively, upregulated and downregulated. In this context, the expression level of all these genes agrees well with the reduction in cancer cell proliferation.

MDM2 is an oncogene that acts as a negative regulator of the tumor-suppressor p53 [40] and it is overexpressed in many tumors [41]. Active p53 is known to downregulate the expression of several genes responsible for cancer development and its anticancer role has been widely described [42]. JUN is a transcription factor that plays a key role in controlling cell proliferation, apoptosis and differentiation [43], whose amplification has been associated with cancer progression and growth and with the increase in invasive properties [44,45].

The interaction between p53 and JUN is crucial for cell cycle regulation. In the absence of JUN, the basal level of p53 is increased resulting in reduced cell proliferation. On the contrary, JUN overexpression negatively regulates the transcription of p53, blocking the p53-mediated growth suppression [43].

The balance between MDM2 and p53 maintains a healthy cellular state. MDM2 is an E3 ubiquitin ligase that negatively regulates p53 [46], but it can also act in p53-independent ways modulating different pathways [47]. Furthermore, an increased expression of MDM2 is often present in cancer with mutated p53. 

To better analyze if MDM2 could be the relevant causative gene linked to AGS anticancer effects, we verified the efficacy of AGS treatment and the molecular pathway activated in p53-defective cancer cell lines. To this aim, we chose to use two cell lines characterized by high aggressive behavior and invasiveness: MDA, a breast cancer cell line that lacks of functional p53 due to a missense mutation [25] and PC3, a prostate cancer cell line with a mutation in the p53 gene that results in a truncated protein [26].

Our results confirmed that AGS treatment in both cell lines determined an increase in apoptosis, while no effect was evidenced in the corresponding normal non-tumor cell lines, MCF10A and PNT2. q-PCR analysis confirmed, in both cell lines, the expression levels of the genes in the network involved in apoptosis. Protein expression analysis clearly showed, in both cell lines, a decreased expression of JUN and MDM2, thus reinforcing the possibility that MDM2 could be the molecular target of AGS treatment. Finally, to understand the signaling pathway activated in the absence of p53 and involved in MDA and PC3 cell death, we analyzed the protein expression of p27 and p21, two CDK inhibitors whose upregulation has been reported to inhibit cell proliferation in both cancers. p21 and p27 have been described to bind MDM2 and activate apoptosis also in p53-defective contexts [29,30,31]. 

Our results confirmed that MDM2 downregulation is linked to upregulation of p21 and p27 expression in breast cancer cells. On the contrary, in PC3, decreased MDM2 appears to be linked to the exclusive upregulation of p21. These results suggest a p21 pro-apoptotic role in the absence of p53.

In order to gain insights about the biochemical changes occurring in the cancer cells following AGS treatment, we carried out a metabolomics analysis either at “global” untargeted or targeted levels. Cancer cells are characterized by metabolic changes that are needed for their proliferation and different metabolic rearrangements which are involved in cancer growth [48]. In fact, increased cell proliferation involves changes in energy metabolism and nutrient uptake. On the other hand, the metabolic shift in cancer can be influenced by oncogenes and tumor suppressors. More specifically, cancer metabolism is characterized by high glucose/energy metabolism that is used to produce energy (Warburg effect). Other metabolic changes involve the pentose phosphate pathway (PPP) as well as biosynthesis of lipids, proteins and nucleic acids and TCA [48].

In our study, metabolomic analysis of MSTO, PC3 and MDA cells following AGS treatment detected 115 DAMs. Interestingly, in all the cell lines analyzed, we observed a metabolic shift with a reduction in lactic acid, the glucose fermentation product, and an increase in the intermediates of the citric acid pathway, such as citric acid, isocitric acid and α-ketoglutaric acid. These data suggest an inhibition of the Warburg effect and an activation of the mitochondrial activity, resulting in apoptosis. 

Notably, PC3 and MDA cell lines displayed a more extreme attitude compared to MSTO, as revealed both by PCA and HCL analysis, as well as by the number of common and DAMs metabolites. This aspect could reflect the lack of functional p53.

Using a metabolomics annotation pipeline and focusing on ions with a consistent trend in at least 2 out of 3 cell lines, we were able to assign 37 chemical formulas and 25 identities. In most of the cases, metabolic alterations agree well with the decreased tolerance to cancer induction and proliferation. A notable example is the increase in all cell lines of Alpha-CEHC, produced in the frame of tocopherol catabolism that has been described to have anticancer properties [49]. 

We found changes in metabolites, belonging to a lipid class, that have been described to have anticancer properties (e.g. eicosatrienoic acids) [50] or to be associated with low risk of cancer development (as ( + )-16-methyl stearic acid) [51]. Furthermore, we found a group of phospholipids with higher contents in AGS-treated cells. These molecules can have a dual role since their overexpression can be associated with cell growth increase, while others might show opposite functions, determining growth decrease and apoptosis increase [52].

We also observed that AGS was able to modulate metabolites belonging to the amino acid class. It has been described that MDM2, independently of p53, is able to transcriptionally regulate amino acid metabolism and, in particular, act on serine/glycine metabolism contributing to cancer cells’ anabolic demand [53]. Our data support this observation, since we noticed a reduced amount of serine. Moreover, we found L-carnitine reduced in MSTO and MDA cells. This is a very interesting finding since the carnitine system is known to have a central role in cancer metabolic plasticity. Indeed, it interplays with key factors that supply the increased energetic and biosynthetic demand of cancer cells [54].

Other metabolic changes consistent with AGS anti-cancer activity were found in folic acid [55] as well as in benzoic acid and its derivatives [56]. Finally, AGS-treated cells showed a reduced content in some N compounds with primary roles for cell metabolism. This aspect is of particular interest since it is known that cancer cells need high amounts of nitrogen [57].

Our functional enrichment analysis highlighted the involvement of several metabolic pathways modulated by AGS treatment: Warburg effect, pyruvate/glycolysis and gluconeogenesis. All these pathways have been reported to play a key role in cancer [48].

Overall, these results agree well with previous studies, indicating that activation of p53 affects cancer properties by modulating the glucose metabolism and the Warburg effect through inhibition of MDM2 [58]. Furthermore, our data suggest that MDM2 could modulate glucose metabolism also independently of p53 [59].

To validate, *in vivo*, the AGS anticancer effects, we generated a xenograft mesothelioma mouse model. The results indicated that administration of AGS three times/week for up to four weeks strongly delayed tumor growth, when compared to control, untreated tumors. According to this observation, immunohistochemical analysis indicated that tumor samples from AGS-treated mice showed a decrease in proliferation index and an increase in apoptosis. In addition, we also confirmed *in vivo* that AGS determined a reduction in MDM2 expression, thus suggesting that MDM2 could represent the causative gene of the apoptosis induced by AGS treatment. 

## 5. Conclusions 

Combined transcriptomics and metabolomics analysis allowed us to unravel the unknown mechanism of action of AGS against mesothelioma, breast and prostate cancer. Overall, our results indicate that apoptosis induced by AGS-treatment occurred through MDM2 downregulation and concomitant modulation of glucose metabolism. *In vivo* analysis confirmed the downregulation of MDM2 and the concomitant activation of the apoptotic pathway, suggesting that MDM2 can be viewed as a candidate target to induce an anticancer effect. Notably, MDM2 has been shown to act as oncogene in human cancers, and different strategies to inhibit MDM2 are under analysis for their treatment. The promising results of our study rely on the efficacy of AGS treatment also in p53-defective cancer cells and on the finding that MDM2 represents the key mediator of AGS activity. Taken together, our data suggest that AGS could be viewed as a natural source for selecting novel anticancer drug molecules exerting pro-apoptotic activity in different cancers. 

## Figures and Tables

**Figure 1 cells-12-00210-f001:**
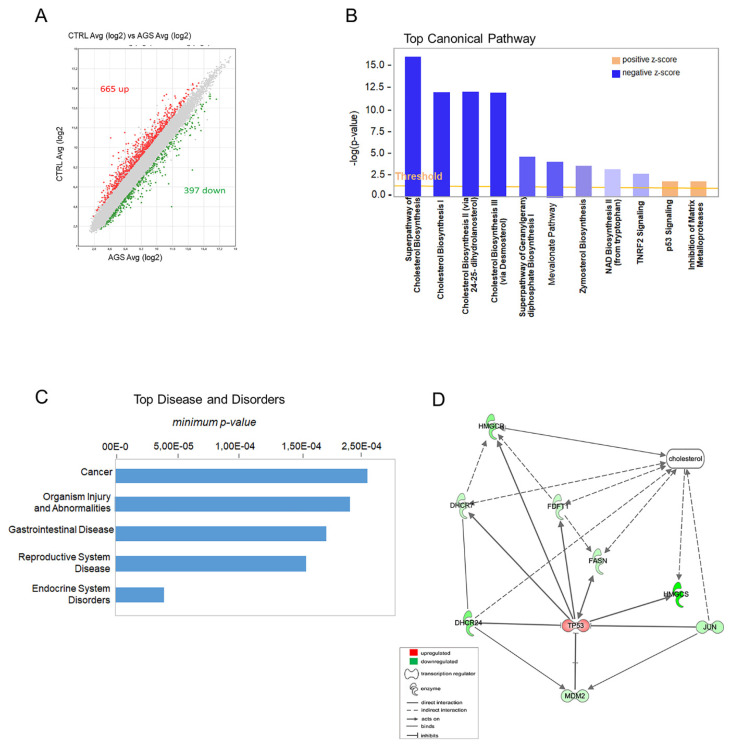
Functional annotation of AGS deregulated genes. (**A**) AGS treatment detected about one thousand deregulated genes: 665 upregulated and 397 downregulated. (**B**) IPA top enriched canonical pathways to which belong deregulated genes. (**C**) IPA classification of the most representative Diseases and Disorders. (**D**) IPA functional network analysis that associated the genes of the cholesterol biosynthesis and genes involved in apoptosis.

**Figure 2 cells-12-00210-f002:**
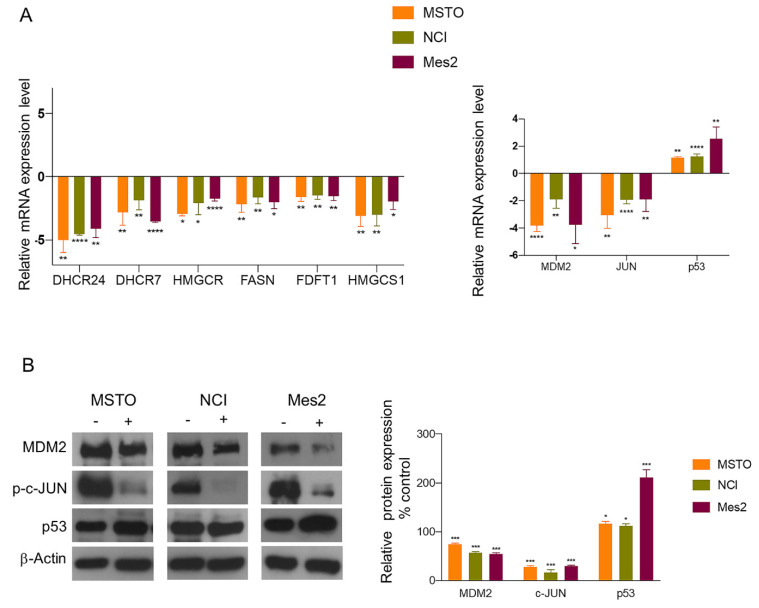
Biological validation of the AGS-deregulated genes included in the network. (**A**) q-PCR analysis of the expression of genes involved in the cholesterol biosynthesis and in apoptosis in MSTO, NCI and Mes2 cells. (**B**) Protein analysis of the genes involved in apoptosis, MDM2, JUN and p53 in MSTO, NCI and Mes2 cells after AGS treatment. The relative expression levels were compared to the β-actin used as loading control. (-): untreated cells, (+): AGS- treated cells. Data are presented as the mean ± standard deviation (*n* = 3). * *p* < 0.05, ** *p* < 0.01, *** *p* < 0.0004, **** *p* < 0.0001 vs. control. All data are representative of at least three independent experiments.

**Figure 3 cells-12-00210-f003:**
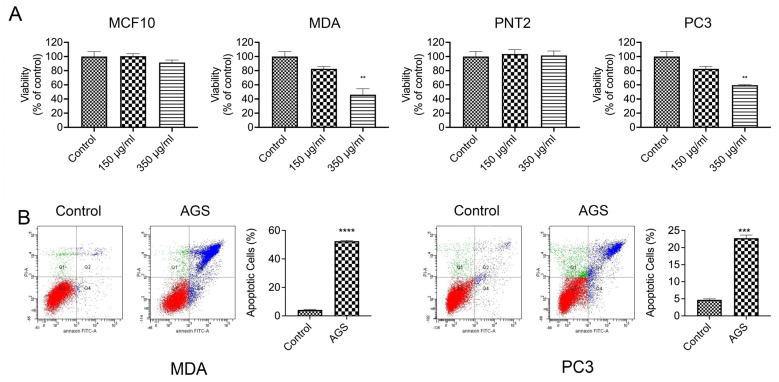
AGS treatment induces apoptosis in breast and prostate cancer, MDA and PC3. (**A**) AGS determines cell viability decrease in MDA and in PC3 cells, while it does not have effect on the non-tumoral cells MCF10A and PNT2. (**B**) FACS analysis showed that AGS treatment triggers apoptosis in both cell lines with a major effect in MDA, as reported by the histograms. Data are presented as the mean ± standard deviation (*n* = 3). ** *p* < 0.01, *** *p* < 0.0004, **** *p* < 0.0001 vs. control. Control indicates untreated cells. All data are representative of at least three independent experiments.

**Figure 4 cells-12-00210-f004:**
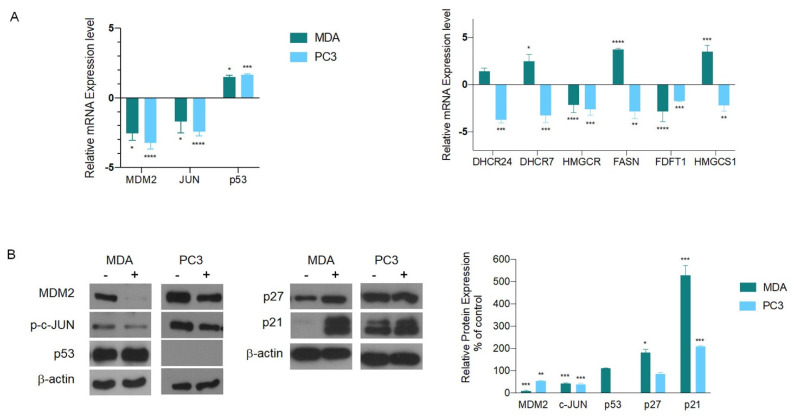
Biological validation of the genes included in the network in MDA and PC3 after AGS treatment. (**A**) q-PCR analysis of the expression of genes involved in apoptosis and in the cholesterol biosynthesis. (**B**) Protein analysis of the genes involved in apoptosis, MDM2, JUN and p53 and of p21 and p27, two MDM2 targets. The relative expression levels compared to the b-actin used as loading control. (-): untreated cells, (+) AGS- treated cells. Data are presented as the mean ± standard deviation (*n* = 3). * *p* < 0.05, ** *p* < 0.01, *** *p* < 0.0004, **** *p* < 0.0001 vs. control. All data are representative of at least three independent experiments.

**Figure 5 cells-12-00210-f005:**
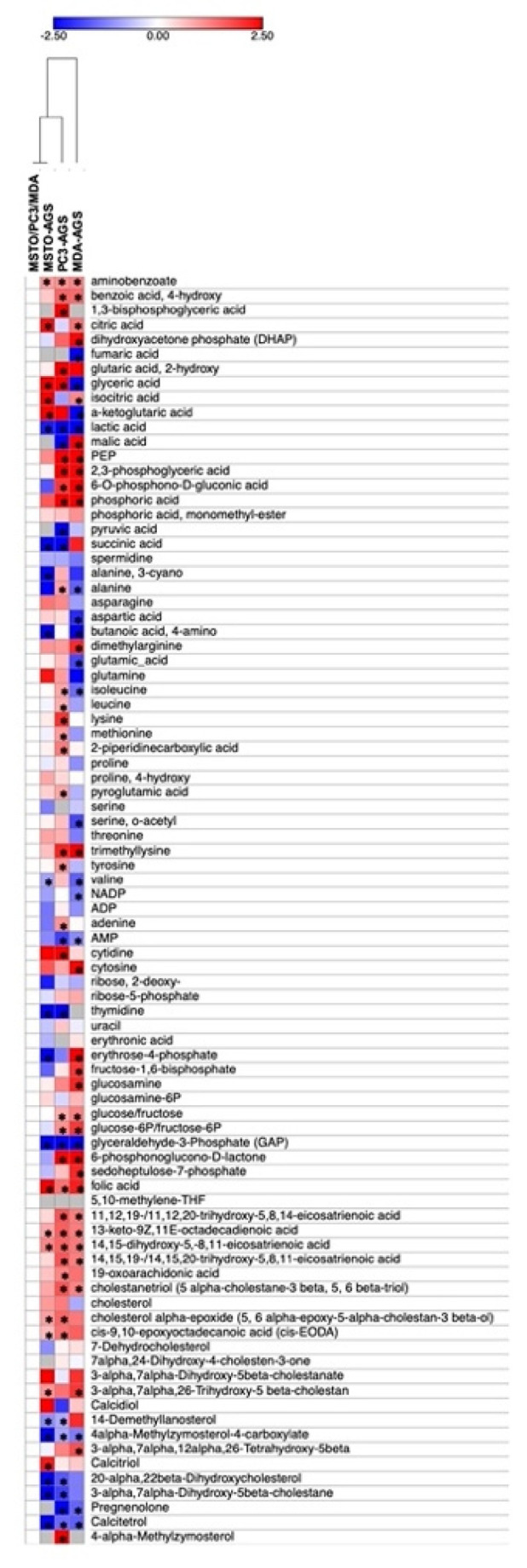
Hierarchical Clustering (HCL) of targeted metabolome of AGS-treated tumor cell lines. Data were normalized, for each cell line, to their respectively untreated control samples and expressed as log2. Asterisks indicate differentially accumulated metabolites (DAMs) according to an ANOVA + Tukey’s pairwise *t*-test. All data are representative of at least three independent experiments.

**Figure 6 cells-12-00210-f006:**
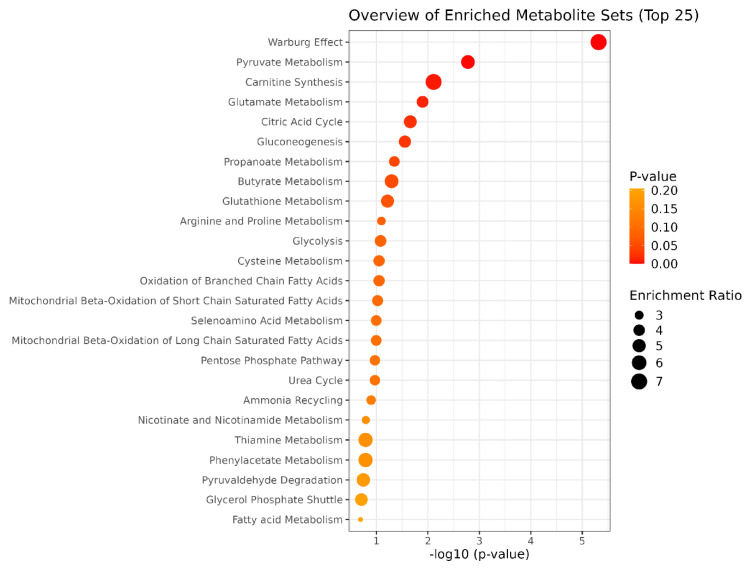
Functional enrichment analysis of differentially accumulated metabolites. Dot Plots of pathway enrichment analysis of metabolites using the MetabolAnalysist5.0 tool (for more details, see materials and methods). Only metabolites altered in at least two out of three cell lines were considered.

**Figure 7 cells-12-00210-f007:**
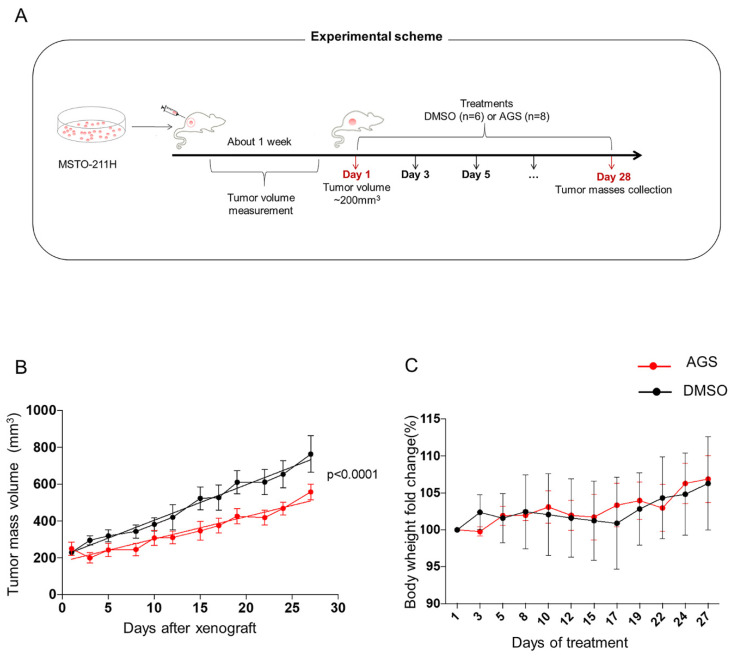
*In vivo* anticancer properties of AGS. (**A**) Experimental scheme showing design and details of the mesothelioma xenograft mouse model (for details see methods). (**B**) AGS administration significantly delayed the growth of the ectopic tumors as compared to the untreated tumors (DMSO) (*p* < 0.0001). (**C**) Mice body weight measured during the entire period of the treatment and expressed as percentage with respect to the first day of treatment. Data are expressed as means ± SEM for each group and the linear regression (dashed line) is reported.

**Figure 8 cells-12-00210-f008:**
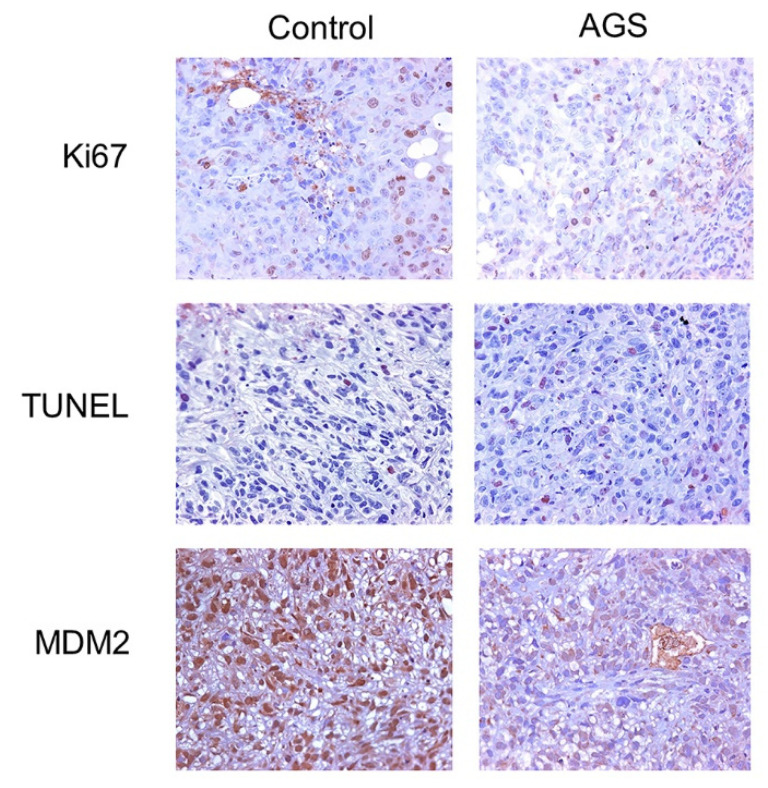
*In vivo* effects of AGS on cell proliferation, apoptosis and on MDM2 expression. Representative images of immunohistochemical staining of tumors from AGS-treated compared to the vehicle treated (Control) (original magnification ×20). AGS treatment determines a significant decrease in cell proliferation (*p* < 0.0004) and increase in apoptosis (*p* < 0.0004) as well as a reduction in MDM2 expression (*p* < 0.0001).

**Table 1 cells-12-00210-t001:** qPCR primers.

Gene	Forward	Reverse
*p53*	5′-TCTGTCCCTTCCCAGAAAACC-3′	5′-CAAGAAGCCCAGACGGAAAC-3′
*JUN*	5′-TTTTGCAAGCCTTTCCTGCG-3′	5′-TCTTCTCTTGCGTGGCTCTC-3′
*MDM2*	5′-AAGGTGGGAGTGATCAAAAGGA-3′	5′-TAGAAACCAAATGTGAAGATGAAGGT-3′
*DHCR24*	5′-ACATCTGCACTGCTTACGAG-3′	5′- AAACCCAGCGTCCCACAG-3′
*DHCR7*	5′- CGCAGGACTTTAGCCGGT-3′	5′-TGGCTTTGGGAATGTTGGGT-3′
*FDFT1*	5′-GGAAGGTGATGCCCAAGATG-3′	5′-ACTGGTCTGATTGAGATACTTGTAGCA-3′
*HMGCR*	5′-CCTTTCCAGAGCAAGCACATTA-3′	5′-TTTCCCTTACTTCATCCTGTGAGTT-3′
*HMGCS1*	5′-TGCTGTCTTCAATGCTGTTAACTG-3′	5′-ACCAGGGCATACCGTCCAT-3′
*GAPDH*	5′-CAAGGCTGTGGGCAAGGT-3′	5′-GGAAGGCCATGCCAGTGA-3′

**Table 2 cells-12-00210-t002:** Bioprofiler analysis of AGS deregulated genes identifying *MDM2* as causative gene.

MDM2	Effect on Disease	Disease or Function
decreased	affects	Aberration of Chromosomes, Abnormal cell cycle
decreased	affects	Abnormal cell cycle
decreased	increases	Apoptosis
decreased	increases	Apoptosis of lymphocytes
decreased	decreases	Cell viability of lumphoma cell lines
decreased	affects	Abnormal morphologyof granule neuron progenitors cells
decreased	increases	Arrest in G1 phase of lung cancer cell lines
increased	decreases	Apoptosis of hematopoietic cell lines
increased	affects	Morphology of bone cancer and sarcoma cell lines
increased	decreases	Stabilization of chromosomes
increased	decreases	Arrest in G1 phase of bone cancer cell lines
increased	increases	Invasion of tumor cell lines
increased	increases	Cancers and tumors
increased	increases	Arrest in cell cycle progression
increased	increases	Proliferation of cells
increased	decreases	Apoptosis
increased	increases	Development of mammary tumor
increased	increases	Lung metastasis by tumor
increased	decreases	Arrest in proliferation of cells

**Table 3 cells-12-00210-t003:** Differentially accumulated metabolites following AGS treatments. Up and down arrows represent over- and under-accumulated metabolites.

MSTO	PC3	MDA
14-Demethyllanosterol ↓	1,3-bisphosphoglyceric acid ↑	11,12,19-/11,12,20-trihydroxy-5,8,14-eicosatrienoic acid ↑
20-alpha,22beta-Dihydroxycholesterol ↓	11,12,19-/11,12,20-trihydroxy-5,8,14-eicosatrienoic acid ↑	13-keto-9Z,11E-octadecadienoic acid ↑
3-alpha,7alpha-Dihydroxy-5beta-cholestane ↓	13-keto-9Z,11E-octadecadienoic acid ↑	14,15-dihydroxy-5,-8,11-eicosatrienoic acid ↑
3-alpha,7alpha,26-Trihydroxy-5 beta-cholestan ↑	14-Demethyllanosterol ↓	14,15,19-/14,15,20-trihydroxy-5,8,11-eicosatrienoic acid ↑
4alpha-Methylzymosterol-4-carboxylate ↓	14,15-dihydroxy-5,-8,11-eicosatrienoic acid ↑	2,3-phosphoglyceric acid ↑
a-ketoglutaric acid ↑	14,15,19-/14,15,20-trihydroxy-5,8,11-eicosatrienoic acid ↑	3-alpha,7alpha,12alpha,26-Tetrahydroxy-5beta ↑
alanine, 3-cyano ↓	19-oxoarachidonic acid ↑	3-alpha,7alpha,26-Trihydroxy-5 beta-cholestan ↑
Aminobenzoate ↑	2-piperidinecarboxylic acid ↑	4alpha-Methylzymosterol-4-carboxylate↓
butanoic acid, 4-amino ↓	2,3-phosphoglyceric acid ↑	6-O-phosphono-D-gluconic acid ↑
Calcitetrol ↓	20-alpha,22beta-Dihydroxycholesterol↓	6-phosphonoglucono-D-lactone ↑
Calcitriol ↑	3-alpha,7alpha-Dihydroxy-5beta-cholestane ↓	a-ketoglutaric acid ↓
cholesterol alpha-epoxide (5, 6 alpha-epoxy-5-alpha-cholestan-3 beta-ol) ↑	4-alpha-Methylzymosterol ↑	Alanine ↓
cis-9,10-epoxyoctadecanoic acid (cis-EODA) ↑	4alpha-Methylzymosterol-4-carboxylate ↓	Aminobenzoate ↑
citric acid ↑	6-O-phosphono-D-gluconic acid ↑	AMP ↓
erythrose-4-phosphate ↓	6-phosphonoglucono-D-lactone ↑	aspartic acid ↓
folic acid ↑	Adenine ↑	benzoic acid, 4-hydroxy ↑
glyceraldehyde-3-Phosphate (GAP) ↓	Alanine ↑	butanoic acid, 4-amino ↓
glyceric acid ↑	Aminobenzoate ↑	Calcitetrol ↓
isocitric acid ↑	AMP ↓	cholestanetriol (5 alpha-cholestane-3 beta, 5, 6 beta-triol) ↑
lactic acid ↓	benzoic acid, 4-hydroxy ↑	citric acid ↑
succinic acid ↓	Calcitetrol ↓	Cytosine ↑
Thymidine ↓	cholestanetriol (5 alpha-cholestane-3 beta, 5, 6 beta-triol) ↑	dihydroxyacetone phosphate ↑
Valine ↓	cholesterol alpha-epoxide (5, 6 alpha-epoxy-5-alpha-cholestan-3 beta-ol) ↑	Dimethylarginine ↑
	cis-9,10-epoxyoctadecanoic acid (cis-EODA) ↑	erythrose-4-phosphate ↑
	Cytidine ↑	folic acid ↑
	folic acid ↑	fructose-1,6-bisphosphate ↑
	glucose-6P/fructose-6P ↑	fumaric acid ↓
	glucose/fructose ↑	Glucosamine ↑
	glutaric acid, 2-hydroxy ↑	glucose-6P/fructose-6P ↑
	glyceraldehyde-3-Phosphate (GAP) ↓	glucose/fructose ↑
	glyceric acid ↑	glutamic acid ↓
	Isoleucine ↑	glyceraldehyde-3-Phosphate (GAP) ↓
	lactic acid	glyceric acid ↓
	Leucine ↑	isocitric acid ↑
	Lysine ↑	Isoleucine ↓
	malic acid ↓	lactic acid ↓
	Methionine ↑	malic acid ↑
	PEP ↑	NADP ↓
	phosphoric acid ↑	PEP ↑
	Pregnenolone ↓	phosphoric acid ↑
	pyroglutamic acid ↑	Pregnenolone ↓
	pyruvic acid ↓	sedoheptulose-7-phosphate ↑
	succinic acid ↓	serine, o-acetyl ↓
	Thymidine ↓	Trimethyllysine ↑
	Trimethyllysine ↑	Valine ↓
	Tyrosine ↑

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
