# Peer review of "Aglianico Grape Seed Semi-Polar Extract Exerts Anticancer Effects by Modulating MDM2 Expression and Metabolic Pathways"

_cells, 2023, doi:10.3390/cells12020210_

Round 1

Reviewer 1 Report

The manuscript requires the following corrections.

In Abstract, line no. 26, …. resulted enriched in oligomeric…. It should be: ……resulted in enriched oligomeric….

In Abstract, line no.32 …AGS significantly inhibited tumor progression in a xenograft a mouse model of mesothelioma. Grammatically incorrect.

Line no. 34-35……can be viewed as a source to select novel anticancer drug…….  Reframe the sentence to make it simple and clear.

Line no. 43… Sentence should start with a capital letter.

Line no. 48: …key molecules responsible of wide biological functions… See if ‘of’ can be replaced with ‘for’.

Line no. 55: bio-active can be written as ‘bioactive’

Line no. 60: These properties are linked to the peculiar ability of a specific proanthocyanidins……Please be mindful of singular and plural.

In line no. 101, why the word briefly is written?

In line no. 130…do shorten the make/address.

In line no. 154…. Life Technologies belongs to …..complete it. (Life Technologies, country).

Line no. 159-177, do tabulate this information.

In Line no. 178-179, do mention the country… (Applied Biosystems, country).

Line no. 431 and 432 need to be corrected.

Line no. 436….it should be ‘suppressor’ and not suppressors.

Line no. 441…  “On the contrary, p27 resulted not modulated in PC3” …… Reframe the sentence.

Line no. 445…. Instead of ‘hints’…it can be ‘indicates’.

Line no. 468-469…… check the sentence framing.

Line no. 472-474…Please write the sentence in direct speech. It seems some plagiarism removal tool has been used to frame sentences.

Figure no. 5 (line no 481)… on untreated control samples according the comparison…… (correct it).

Line no. 495…biosynthesis resulted down-accumulated: for instance…(not clear…check it).

Line no. 504…which resulted over-accumulated in PC3….. (correct it).

Line no. 518…belonging to N compound group, resulted reduced, respectively… (reframe the statement for better understanding).

In table 2,  colour coding is not required. Kindly use ↑ or ↓

Line no. 539, ….AGS treatment Additionally…(There should be a full-stop in between).

Line no. 599, …. against several cancer [38] ….. (It should be cancers).

Line no. 673,….. the mitochondria activity….(It should be mitochondrial activity).

Line no. 701, showed a reduced contents in some N compounds…(It should be ‘content’)

Line no. 721…mention here also which type of cancer?

Line no. 731….it should be ‘cancers’.

In the reference section: The page numbering is not done in a similar fashion. Eg: see the page numbering of the reference no. 51 and 52.

Reference no. 57, 58, 41, 48, 50, 36, 21, 12, 10, 9, 7, 5, 2…page numbers/range is missing.

Author Response

Point by Point answer

We thank the reviewer for the helpful comments and suggestion.

All the suggested corrections were included in the revised version of the manuscript.

The manuscript requires the following corrections.

In Abstract, line no. 26, …. resulted enriched in oligomeric…. It should be: ……resulted in enriched oligomeric….

In Abstract, line no.32 …AGS significantly inhibited tumor progression in a xenograft a mouse model of mesothelioma. Grammatically incorrect.

Line no. 34-35……can be viewed as a source to select novel anticancer drug…….  Reframe the sentence to make it simple and clear.

Line no. 43… Sentence should start with a capital letter.

Line no. 48: …key molecules responsible of wide biological functions… See if ‘of’ can be replaced with ‘for’.

Line no. 55: bio-active can be written as ‘bioactive’

Line no. 60: These properties are linked to the peculiar ability of a specific proanthocyanidins……Please be mindful of singular and plural.

In line no. 101, why the word briefly is written?

In line no. 130…do shorten the make/address.

In line no. 154…. Life Technologies belongs to …..complete it. (Life Technologies, country).

Line no. 159-177, do tabulate this information.

In Line no. 178-179, do mention the country… (Applied Biosystems, country).

Line no. 431 and 432 need to be corrected.

Line no. 436….it should be ‘suppressor’ and not suppressors.

Line no. 441…  “On the contrary, p27 resulted not modulated in PC3” …… Reframe the sentence.

Line no. 445…. Instead of ‘hints’…it can be ‘indicates’.

Line no. 468-469…… check the sentence framing.

Line no. 472-474…Please write the sentence in direct speech. It seems some plagiarism removal tool has been used to frame sentences.

Figure no. 5 (line no 481)… on untreated control samples according the comparison…… (correct it).

Line no. 495…biosynthesis resulted down-accumulated: for instance…(not clear…check it).

Line no. 504…which resulted over-accumulated in PC3….. (correct it).

Line no. 518…belonging to N compound group, resulted reduced, respectively… (reframe the statement for better understanding).

In table 2,  colour coding is not required. Kindly use ↑ or ↓

Line no. 539, ….AGS treatment Additionally…(There should be a full-stop in between).

Line no. 599, …. against several cancer [38] ….. (It should be cancers).

Line no. 673,….. the mitochondria activity….(It should be mitochondrial activity).

Line no. 701, showed a reduced contents in some N compounds…(It should be ‘content’)

Line no. 721…mention here also which type of cancer?

Line no. 731….it should be ‘cancers’.

Answer: We thank the reviewer for the helpful suggestions. We have inserted in the revised version of the manuscript all the indicated corrections that are highlighted in red.

In the reference section: The page numbering is not done in a similar fashion. Eg: see the page numbering of the reference no. 51 and 52.

Reference no. 57, 58, 41, 48, 50, 36, 21, 12, 10, 9, 7, 5, 2…page numbers/range is missing.

Answer: We have corrected the references where the page number were missing.

Reviewer 2 Report

The present data is interesting and authors have presented data in a understandable and easy format. However, there are several improvements required before the final acceptance and authors should address the concerns.

Language editing is required because some of the sentence’s meanings are missing

Unnecessary use of commas and inconsistency with the use of oxford comma

Typographical errors were observed

Why did the authors choose to give two rounds of AGS treatment instead of just one, please explain.

The authors are suggesting that the AGS treatment increases the protein level of DHCR24 in lines 373-374. However, authors have also presented data in 2A that the AGS decreases the transcript level of DHCR24. Can they explain how treatment can cause low transcript levels but increased protein levels? Also, how DHCR24 regulates p53? If it inhibits p53 then AGS should decrease DHCR24 at the protein level or it can be the case that the expression of AGS-mediated p53 expression is independent of DHCR24. The authors should clearly emphasize these points in the manuscript thoroughly.

What is the correct concentration of AGS treatment? In the text, the authors mention 350ug/ml but the figure shows 300ug/ml. Also, the concentration can either be in ug/ml or uM unit not uM/ml. Please check it carefully. The author should also include the flow cytometry data of normal cells MCF10A and PNT2.

The authors state that MDA and PC3 cells have non-functional p53 due to mutation. Then what is the logic to check its expression under AGS-treated conditions? How the upregulation or downregulation of a non-function gene can have a consequence on cancer development? Please explain.

In line number 429-430, the authors emphasize that there is no change in the expression of the p53 gene in MDA cells. However, in Fig 4A and 4B, the authors are showing transcript and protein level changes in MDA cells having a significance of 0.05 as suggested by *. The correlation between the text and the figure is missing.

The in vivo xenograft experiment was performed using MSTO cells. Can the author explain why they chose not to mention the cell line in the result section? Also, all the work including the mechanism and metabolomics was done using other cell lines. Why the authors suddenly chose to use the MTSO cell line and not the other? The authors are suggested to validate the in vivo data using another cell line too (at least one cell line).

Author Response

Point by Point answer

We thank the reviewer for the helpful comments and suggestion

All the suggested corrections were included in the revised version of the manuscript

The present data is interesting and authors have presented data in a understandable and easy format. However, there are several improvements required before the final acceptance and authors should address the concerns.

Answer: We thank the reviewer for his comments and suggestions. We have corrected the manuscript following the reviewer comments.

Language editing is required because some of the sentence’s meanings are missing

Answer: We have revised thoroughly all the manuscript.

Unnecessary use of commas and inconsistency with the use of oxford comma

Answer: We have eliminated all unnecessary commas.

Typographical errors were observed

Answer: We have corrected typos error.

Why did the authors choose to give two rounds of AGS treatment instead of just one, please explain.

Answer: We have described in our previous work that the best time/dosage treatment that evidenced AGS-induced apoptosis, was a 48-hour treatment in which the extract was added twice, every 24 hours (24+24 hour). This treatment determined the strongest effect on cell viability decrease (Di Meo et al., JFF 2019). This effect might be related to the presence in the extract of active molecules characterized by a short half-life.

The authors are suggesting that the AGS treatment increases the protein level of DHCR24 in lines 373-374. However, authors have also presented data in 2A that the AGS decreases the transcript level of DHCR24. Can they explain how treatment can cause low transcript levels but increased protein levels?

Answer: We have better explained this phenomenon in the text adding the following sentence:

Despite the increased expression of DHCR24 protein level we did not observe a reduction of the expression of p53, suggesting that in these cell lines DHCR24 could be post-translationally regulated as previously reported [23]”.

Also, how DHCR24 regulates p53? If it inhibits p53 then AGS should decrease DHCR24 at the protein level or it can be the case that the expression of AGS-mediated p53 expression is independent of DHCR24. The authors should clearly emphasize these points in the manuscript thoroughly.

Answer: Although as described in the manuscript (lines 329-333):

DHCR24 has been previously reported to reduce the activity of p53, directly by reducing its activation, and indirectly by increasing the interaction between MDM2 and p53, thus leading to the ubiquitination of the latter [17]. Additionally, DHCR24 has been shown to interact with MDM2 independently of p53, possibly affecting other MDM2 targets [18]

our results indicate that DHCR24 is not related to AGS-induced apoptosis, so we did not further investigate this aspect in this study. This phenomenon could be related to other mechanisms activated by AGS in mesothelioma cell lines. We will further investigate on this aspect in the future.

What is the correct concentration of AGS treatment? In the text, the authors mention 350ug/ml but the figure shows 300ug/ml. Also, the concentration can either be in ug/ml or uM unit not uM/ml. Please check it carefully. The author should also include the flow cytometry data of normal cells MCF10A and PNT2.

Answer: We thank the reviewer for his observation. The right concentration of AGS used was 350ug/ml and we have corrected it in the figure accordingly.

Regarding FACS analysis on MCF10A and PNT2 cells we had already performed this experiment, but we do not included the result in the manuscript because it does not add value to the paper. In fact, as you can see below, AGS treatment does not affect viability in both these cell lines.

The authors state that MDA and PC3 cells have non-functional p53 due to mutation. Then what is the logic to check its expression under AGS-treated conditions? How the upregulation or downregulation of a non-function gene can have a consequence on cancer development? Please explain.

Answer: We performed this experiment to further verify that AGS was able to induce apoptosis independently of p53 and acting specifically on MDM2, confirming that MDM2, represents the key player in the apoptosis induced by AGS.

In line number 429-430, the authors emphasize that there is no change in the expression of the p53 gene in MDA cells. However, in Fig 4A and 4B, the authors are showing transcript and protein level changes in MDA cells having a significance of 0.05 as suggested by *. The correlation between the text and the figure is missing.

Answer: We thank the reviewer for this comment. The correlation between the text and the Figure 4A and 4B has been corrected. We also corrected the typos error in the histogram of p53 (Figure 4B).

The in vivo xenograft experiment was performed using MSTO cells. Can the author explain why they chose not to mention the cell line in the result section? Also, all the work including the mechanism and metabolomics was done using other cell lines. Why the authors suddenly chose to use the MTSO cell line and not the other? The authors are suggested to validate the in vivo data using another cell line too (at least one cell line).

Answer: We thank the reviewer and have mentioned MSTO in the corresponding section.

For the in vivo experiment we injected only MSTO cells that are the ones tumorigenic in vivo.

In the next study we will plan to verify the AGS in vivo efficacy also in other cancers.

Round 2

Reviewer 2 Report

For this point

"The authors are suggesting that the AGS treatment increases the protein level of DHCR24 in lines 373-374. However, authors have also presented data in 2A that the AGS decreases the transcript level of DHCR24. Can they explain how treatment can cause low transcript levels but increased protein levels?"

I actually wanted to know how AGS treatment can have low DHC24 transcript level but increased protein level. Can you please explain how post-translational (as referred by authors) can increase the protein level with low transcript level.

Author Response

Point by point answer

"The authors are suggesting that the AGS treatment increases the protein level of DHCR24 in lines 373-374. However, authors have also presented data in 2A that the AGS decreases the transcript level of DHCR24. Can they explain how treatment can cause low transcript levels but increased protein levels?"

I actually wanted to know how AGS treatment can have low DHC24 transcript level but increased protein level. Can you please explain how post-translational (as referred by authors) can increase the protein level with low transcript level.

Answer: We thank the reviewer for the comment.

We want to underline that we just have reported the data we found. Since DHCR24 expression is not related to AGS-induced apoptosis, in all the analysed cell lines, we did not further investigate this aspect.

We just can hypothesize that there is a compensatory feedback effect that can stabilizes the cellular mRNA thus resulting in protein increase. This effect has been elsewhere described for other genes (Schlaepfer IR et al., Lipid catabolism via CPT1 as a therapeutic target for prostate cancer. Mol Cancer Ther. 2014 Oct;13(10):2361-71; Bianchi M, et al., A negative feedback mechanism links UBC gene expression to ubiquitin levels by affecting RNA splicing rather than transcription. Sci Rep. 2019 Dec 6;9(1):18556.)

We do not affirm that AGS differently regulate transcription and translation of DHCR24, but we only commented the data obtained.